# Chicken *LEAP2* Level Substantially Changes with Feed Intake and May Be Regulated by CDX4 in Small Intestine

**DOI:** 10.3390/ani12243496

**Published:** 2022-12-11

**Authors:** Xiaotong Zheng, Ziwei Chen, Wuchao Zhuang, Jilong Zhang, Jiaheng He, Yinku Xie, Jianfei Chen

**Affiliations:** School of Biotechnology, Jiangsu University of Science and Technology, Zhenjiang 212100, China

**Keywords:** chicken, *LEAP2*, feed intake, expression profile, regulatory mechanism

## Abstract

**Simple Summary:**

Feed-intake optimization is important for the efficient growth of livestock and poultry, and the identification of factors regulating feed intake is crucial for efficient animal husbandry. Liver-expressed antimicrobial peptide 2 (LEAP2) has been reported as an endogenous antagonist of the growth hormone secretagogue receptor (GHSR) and plays an important role in feed intake and energy homeostasis in mammals. In chickens, LEAP2 function is thought to be mainly antimicrobial, and its role in appetite regulation remains unexplored. This study aimed to outline the expression profiles of genes related to the ghrelin system in 20 different tissues of broiler chicks in different energy states. The expression levels of *LEAP2* in the liver and small intestine varied significantly with changes in diet, and CDX4 may be a potential regulator of *LEAP2* expression in the small intestine. Overall, the results of this study highlight the importance of LEAP2 for avian feed intake.

**Abstract:**

Ghrelin O-acyltransferase (GOAT), ghrelin, and GHSR have been reported to play important roles that influence feed intake in mammals. LEAP2, an endogenous antagonist of GHSR, plays an important role in the regulation of feed intake. However, chicken ghrelin has also been reported to have an inhibitory effect on feed intake. The role of the GOAT–Ghrelin–GHSR–LEAP2 axis in chicken-feed intake remains unclear. Therefore, it is necessary to systematically evaluate the changes in the tissue expression levels of these genes under different energy states. In this study, broiler chicks in different energy states were subjected to starvation and feeding, and relevant gene expression levels were measured using quantitative real-time PCR. Different energy states significantly modulated the expression levels of *LEAP2* and *GHSR* but did not significantly affect the expression levels of *GOAT* and *ghrelin*. A high expression level of *LEAP2* was detected in the liver and the whole small intestine. Compared to the fed group, the fasted chicks showed significantly reduced *LEAP2* expression levels in the liver and the small intestine; 2 h after being refed, the *LEAP2* expression of the fasted chicks returned to the level of the fed group. Transcription factor prediction and results of a dual luciferase assay indicated that the transcription factor CDX4 binds to the *LEAP2* promoter region and positively regulates its expression. High expression levels of *GHSR* were detected in the hypothalamus and pituitary. Moreover, we detected *GHSR* highly expressed in the jejunum—this finding has not been previously reported. Thus, GHSR may regulate intestinal motility, and this aspect needs further investigation. In conclusion, this study revealed the function of chicken LEAP2 as a potential feed-intake regulator and identified the potential mechanism governing its intestine-specific expression. Our study lays the foundations for future studies on avian feed-intake regulation.

## 1. Introduction

Feed intake is a limiting factor for poultry-production efficiency, especially in broilers. Ghrelin is a 28-amino-acid peptide hormone that is mainly secreted by the stomach [1]. It is a ligand for the growth hormone secretagogue receptor (GHSR), which plays a critical role in controlling growth hormone release and energy homeostasis [2,3]. The post-translational addition of n-octanoic acid, n-decanoic acid, or an unsaturated form of these fatty acids by ghrelin O-acyltransferase (GOAT) at Ser3 results in acyl-ghrelin, which is necessary for ghrelin to activate GHSR [4,5,6,7]. In mammals, ghrelin is known to stimulate feed consumption, and its plasma concentration has been observed to increase prior to a meal and to decrease after eating [8]. However, in avian species, feed intake is suppressed when ghrelin concentrations increase in the bloodstream [9,10]. In addition, intracerebroventricular administration of ghrelin inhibits feed intake in birds [9,11]. However, peripheral administration of ghrelin at different doses has yielded contradictory results [12,13]. Therefore, the effect of ghrelin in birds seems to be influenced by the route of administration, the dose, and the form of ghrelin used (i.e., acetylated or non-acetylated ghrelin). Overall, ghrelin is considered to be a regulator of feed intake in birds, the mechanisms of which are distinct from those of mammals.

LEAP2 was first reported as an antimicrobial peptide. It exerts antibacterial effects in vitro (e.g., against *Salmonella enteritidis* strains) [14,15]. However, recently, Ge et al. (2018) [16] reported that LEAP2 is an endogenous antagonist of GHSR. Their study indicated that LEAP2 is a peptide hormone that has some appetite-related function [17]. LEAP2 levels have been confirmed to change with feed intake in humans and rats [18], and LEAP2 is highly expressed in the small intestine and the liver [19,20]. A mouse model of vertical sleeve gastrectomy, a common bariatric surgery, indicated that the expression of *LEAP2* can significantly increase in the stomach of this model [16]. LEAP2 has conserved peptide sequences in various animals [21]. However, chicken LEAP2 has been studied as an antibacterial peptide [22,23,24,25,26,27]. As the gastrointestinal system of birds is different from that of humans and rats, it is unclear whether avian LEAP2 acts as a hormone in response to changes in diet [17].

Chicken *ghrelin* and *GHSR* have been cloned and characterized [28,29], and research on the gene expression profiles of *ghrelin* and *GHSR* in broilers indicated that ghrelin serves as a signal of energy utilization and is involved in maintaining energy homeostasis [30]. Furthermore, chicken *ghrelin* and *GHSR* gene polymorphisms are closely related to growth rate [31,32]. *GOAT*, *ghrelin*, and *GHSR* expression profiles in chickens of different ages have also been studied [33]. However, there have been no systematic studies of gene expression before and after fasting.

In summary, chicken *GOAT*, *ghrelin*, *GHSR*, and *LEAP2* systems (especially *LEAP2*) in chickens have not been fully researched in response to different energy states. In this study, *GOAT*, *ghrelin*, *GHSR*, and *LEAP2* levels were evaluated in broiler chicks. First, broiler chicks with different energy states (fed, fasted, and refed groups) were constructed by starvation and refeeding methods, and then the differences in the expression levels of *GOAT*, *ghrelin*, *GHSR*, and *LEAP2* in various tissues among the treatment groups were determined. Finally, we combined the characteristics of the gene expression profiles to explore the potential mechanisms regulating chicken *LEAP2*.

## 2. Materials and Methods

### 2.1. Animals and Sample Collection

Animal protocols were approved by the Institutional Animal Care and Use Committee (IACUC) of Jiangsu University of Science and Technology (G2022SJ13, Zhenjiang, China). Animal care and handling were performed according to the IACUC guidelines.

Male AA-line broiler chicks were purchased from a commercial market for this study. Eighteen 15-day-old broiler chicks were divided into three groups. Each group had six chicks with different treatments. All chicks were housed in the same environmental conditions with free access to water. Commercial complete chick feed was used for raising the chicks. The fed group was fed ad libitum for 18 h; the fasted group was starved for 18 h; the refed group was fasted for 18 h and refed for 2 h. After 18 h, samples were collected, except for the refed group (the refed group chicks needed another 2 h to be fed). Sixteen-day-old chicks were euthanized by anesthesia with CO_2_ and we collected whole blood from the heart from each chick. About 2 mL blood was obtained and preserved in an EP tube, then the blood was naturally coagulated at room temperature for 20 min and centrifuged at 5000× *g* (Eppendorf, Hamburg, Germany) for 10 min to remove the clot. The serum (supernatant) was aliquoted and stored at −80 °C until further use. Tissue samples were collected from the brain, hypothalamus, pituitary, heart, liver, spleen, lung, kidney, gall bladder, crop, proventriculus, gizzard, duodenum, jejunum, ileum, cecum, colon, pectoralis, abdominal skin, and tongue of the birds. All samples were immediately placed in liquid nitrogen after collection and transferred into a refrigerator at −80 °C.

### 2.2. ELISA for Serum Ghrelin and LEAP2

Commercial ELISA kits for chicken ghrelin and LEAP2 were purchased from Senbeijia Biological Technology Co., Ltd. (Nanjing, China), and ELISA was performed according to the manufacturer’s instructions. Briefly, 10 μL chicken serum samples were incubated at room temperature for 20 min, then for 30 min at 37 °C; plates were washed five times in a diluted washing buffer, and 50 μL of secondary antibody (HRP–goat–anti-rabbit) was added to the wells and incubated for 30 min at 37 °C. Excess secondary antibody was washed five times, followed by the addition of a KPL peroxidase substrate solution, which was incubated until a blue color was apparent. The stop solution was added to the wells, and the plates were read using a full wavelength microplate reader (TECAN, Männedorf Switzerland) at 450 nm. Standard curves were generated and used to calculate the serum concentrations of ghrelin and LEAP2.

### 2.3. RNA Isolation and Quantitative Real-Time PCR

All tissue samples were subjected to quantitative real-time PCR (q-PCR) analysis. Total RNA was extracted using RNAiso Plus reagent (Vazyme Biotech Co., Ltd., Nanjing, China). cDNA was synthesized from 1 μg of the extracted total RNA using the PrimerScript RT Reagent Kit with gDNA Eraser (TaKaRa Biomedical Technology Co., Ltd., Beijing, China), as per the manufacturer’s guidelines. q-PCR was performed using a Bio-Rad Light Cycler 96 Real-Time PCR system with 20 μL reaction volumes containing 1 μL cDNA, 10 μL NovoStart SYBR q-PCR SuperMix Plus (Novoprotein Scientific Inc., Suzhou, China), 1 μL each of forward and reverse primers (10 μM), and 7 μL of deionized water. All q-PCR gene-specific primers were designed using Primer Premier 5.0 software. The q-PCR amplification procedure was as follows: 95 °C for 15 min, 40 cycles of 95 °C for 10 s, 58 °C for 20 s, 72 °C for 30 s, and an extension for 10 min at 72 °C. All reactions were performed in triplicate. After a series of tests of the commonly used housekeeping genes, *β-Actin*, *16sRNA*, *GAPDH*, *UB*, and *HPRT*, our pre-experiment revealed that the expression of the housekeeping genes *GAPDH* and *HPRT* was much more consistent than that of others across all tissues; therefore, *GAPDH* and *HPRT* were selected to conduct tissue expression profile experiments in this study. The primers used are listed in Table 1.

### 2.4. Vector Construction

An approximately 1 Kb upstream region from the transcription start site of the chicken LEAP2 gene was cloned using PrimerSTAR GXL DNA polymerase (TaKaRa) and ligated into the pMD19-T vector (TaKaRa), as per the manufacturer’s guidelines. After sequencing by the Sanger method (Shangya, Hangzhou, China), the promoter region of chicken LEAP2 was cloned using PrimerSTAR GXL DNA polymerase with primers containing specific restriction sites (KpN I, Hind III) and then inserted into the multiple copy region of the pGL3.Basic vector (Promega, Beijing, China). A primer sequence mutation method was used to construct the mutated vector pGL3-pro-LEAP2-M (caudal type homeobox 4 (CDX4) binding site mutation plasmid). After Sanger sequencing to ensure the correct sequence, the constructed plasmids were extracted using the EndoFree Mini Plasmid Kit (Tiangen Biotech Co., Ltd., Beijing, China), as per the manufacturer’s guidelines. The CDX4 overexpression plasmid was also constructed as follows: first, the CDS sequence of the CDX4 gene was cloned from the cDNA of small intestine by PrimerSTAR GXL DNA polymerase and ligated into the pMD19-T vector, via a standard cloning method, as described above; then, the PCR products were digested with KpN I and EcoR I enzymes and inserted into a pcDNA3.1 vector to construct CDX4 overexpression vector pcDNA3.1-CDX4. To amplify the target band more efficiently, nested PCR was used in this study. All primers used for cloning the promoter region of LEAP2 and CDX4 are listed in Appendix A, respectively.

### 2.5. Cell Culture and Dual Luciferase Assay

HEK293T cells, which were purchased from ATCC and stored in our laboratory, were used to perform a luciferase assay to identify the promoter region of the chicken LEAP2 gene. Cells were cultured in Dulbecco’s Modified Eagle’s Medium (DMEM) supplemented with 10% fetal bovine serum and penicillin/streptomycin. Cells were cultured at 37 °C and 90% air humidity, with 5% CO_2_. Reagents used for cell cultures, such as fetal bovine serum, DMEM, penicillin/streptomycin, and trypsin, were all purchased from Gibco (Beijing, China). HEK293T cells in good growth conditions were dissociated with 0.25% trypsin and seeded in 24-well plates (Corning Incorporated, New York, USA) in 500 μL with a seeding rate of approximately 50%. When cells reached approximately 70% confluency, they were used for transfection. Six treatments were designed: 0.5 μg pGL3.Basic + 0.5 μg pcDNA3.1, 0.5 μg pGL3-pro-LEAP2 + 0.5 μg pcDNA3.1, 0.5 μg pGL3-pro-LEAP2 + 0.5 μg pcDNA3.1-CDX4, 0.5 μg pGL3-pro-LEAP2 + 1 μg pcDNA3.1-CDX4, 0.5 μg pGL3-pro-LEAP2 + 1.5 μg pcDNA3.1-CDX4, and 0.5 μg pGL3-pro-LEAP2-M + 1 μg pcDNA3.1-CDX4. They were co-transfected with 50 ng pRL-TK using NEOFECT DNA transfection reagent (Neofect biotech Co., Ltd., Beijing, China), according to the manufacturer’s guidelines. The treatments were replicated three times and the assays were technically repeated three times. Forty h after the transfection, the cells were lysed, and then the Firefly and Renilla substrates were added using the Dual Luciferase Reporter Assay Kit (Vazyme), following the manufacturer’s recommendations. Tirefly luciferase signals were normalized to those of Renilla luciferase.

### 2.6. Bioinformatics and Statistical Analysis

The online database Jaspar (http://jaspar.genereg.net/, accessed on 22 February 2022) was used to predict transcriptional factor-binding sites. q-PCR data were collected and analyzed using Bio-Rad CFX Manage (Version 3.1). The CT values of the genes were exported to Microsoft Excel (Version 2016), and the relative gene expression levels were calculated using the 2^−∆∆CT^ method described previously [34]. The genomic mean results of two reference genes (*HPRT* and *GAPDH*) were considered as a normalizer to determine the gene expression profile in this study [35]. The sample mix allowed comparison of the gene expression levels of all samples. The relative expression levels of the other two genes were calculated using the same formula. A two-sided Student’s *t*-test was performed to evaluate the statistical significance of the differences between different groups. One-way ANOVA was conducted using GraphPad Prism 8.0.1 to evaluate the statistial significance of the differences among different groups, and differences were considered significant at *p* ≤ 0.05 or highly significant at *p* ≤ 0.01.

## 3. Results

### 3.1. Tissue Expression Profiles of GOAT, Ghrelin, and GHSR in Chicks

Analysis of *GOAT*, *ghrelin*, and *GHSR* expression patterns in the 20 tested tissues from broiler chicks with different energy states (fed, fasted, and refed) revealed the main tissues expressing these genes and evaluated their response to dietary status. First, *GOAT* was highly expressed in the proventriculus, which is an organ similar with the mammalian stomach, and the hypothalamus had almost the same expression level as the proventriculus; the fed status did not alter the gene expression of *GOAT* (Figure 1A). *Ghrelin* was highly expressed in the proventriculus, and had a moderate expression in the small intestine (duodenum and jejunum), abdominal skin, and lung tissue., The other tissues all had a very low expression level of *ghrelin*, which was almost undetectable. No significant differences were detected in the expression level of *ghrelin* in any tissue among the different experimental groups (Figure 1B). For the *GHSR* gene expression profile, we observed that *GHSR* is a widely expressed gene, with the highest expression level in the advanced central nervous system (including the hypothalamus and the pituitary) and the small intestine (the jejunum only), but it is difficult to detect its expression in the brain and the pectoralis. In total, *GOAT* and *ghrelin* did not respond well to the fed states at the RNA expression level; however, the *GHSR* expression level responded well to the fed states, especially in the jejunum and the liver, while the expression level of the fasted group chickens decreased and the refed group chickens recovered their expression level (Figure 1C).

### 3.2. LEAP2 Changes with Feed Intake in the RNA Expression Level and Serum Content

Analysis of the relative expression of *LEAP2* in the 20 tested tissues from chicks in different energy states revealed significant differences in several tissues among the fed, fasted, and refed groups (Figure 2). Notably, a high expression level of the *LEAP2* gene was detected along the entire small intestine (including the duodenum, jejunum, and ileum) and the liver. Its expression level gradually decreased in the small intestine, from the duodenum to the ileum, exhibiting 0.3- to 0.5-fold lower expression levels than that in the fasted group chicks. Furthermore, *LEAP2* expression levels were 10-fold higher in the livers of fed and refed chicks than in those of fasted chicks. The fed and refed groups showed similar expression levels. In addition, the expression levels of *LEAP2* in the gizzard (*p* = 0.0348), crop (*p* = 0.0476), colon (*p* = 0.0410), and cecum (*p* = 0.0124), which all belong to the digestive tract, were significantly lower in the fasted group than in the fed and refed groups. Furthermore, *LEAP2* had a medium expression level in the kidney and the gall bladder, while the expression levels in other tissues were low.

Our results showed that *LEAP2* expression changed with feed intake at the RNA expression level. However, it is unclear whether LEAP2 circulates. Chicken LEAP2 and ghrelin enzyme-linked immunosorbent assay (ELISA) kits were used to detect serum LEAP2 and ghrelin levels in chicks under different dietary conditions. Serum LEAP2 levels changed with feed intake, while ghrelin levels did not change significantly (Figure 3).

### 3.3. CDX4 Is a Potential Transcriptional Factor of LEAP2 Highly Expressed in the Small Intestine

Chicken *LEAP2* was highly expressed in the whole intestine, especially in the small intestine. We also noted, in a previous report, that chicken *LEAP2* was specifically expressed in the epithelium of the jejunum [23], which suggested that the *LEAP2* gene may have the same regulator in the small intestine. To further examine potential expression regulators of the *LEAP2* gene, a gut-specific transcriptional factor caudal-type homeobox 2 (CDX2) binding site was identified in the upstream (Appendix A) by an online transcriptional binding-site-prediction method; chicken *CDX4* is the homologous gene of *CDX2*. We then determined the *CDX4* gene expression profile in the chicks. The results showed that *CDX4* was highly expressed in the intestine, especially in the small intestine (duodenum, jejunum, and ileum) (Figure 4A). To determine whether CDX4 directly activates the transcription of *LEAP2*, we constructed a luciferase reporter by inserting the sequence of the chicken *LEAP2* promoter region (~1 Kb) into the pGL3.Basic vector, and the CDX4 binding-site mutated vector was constructed using the same method (Figure 4B). Subsequently, a pcDNA–CDX4 overexpression vector was constructed to perform the dual luciferase assay. We found that the wild-type *LEAP2* promoter group showed higher luciferase activity than the pGL3.Basic vector group in HEK293T cells and significantly higher luciferase activity in the CDX4 overexpression group than in the control group; it also had a *CDX4* gene dosage effect (Figure 4C). When we mutated the CDX4 binding sequence, luciferase activity decreased significantly.

## 4. Discussion

Feed intake guarantees the growth of livestock and poultry, and for animal husbandry, exploring the key feed-intake regulatory factors is of great significance. In recent years, many neuropeptides (including central and peripheral neuropeptides) that regulate feeding behavior have been identified in vertebrates. The central neuropeptides include corticotropin-releasing hormone [36], melanocortins [37,38], glucagon [39], and neuropeptide Y [40,41], as well as the peripheral neuropeptides, including leptin and ghrelin. Recently, chicken leptin has been shown to have no obvious hormonal functions [42,43,44]. Thus, chicken ghrelin may play an important role in regulating feed intake. However, the effect of ghrelin in birds seems to be influenced by the route of administration, the dose, and the form of ghrelin used (i.e., acetylated or non-acetylated ghrelin) [9,10,11,12]. Therefore, it is necessary to evaluate the role of ghrelin in regulating feed intake. *GOAT*, *ghrelin*, *GHSR*, and the newly identified endogenous antagonist gene *LEAP2* all belong to the ghrelin system. In this study, we analyzed the gene expression profiles of chicken *GOAT*, *ghrelin*, *GHSR*, and *LEAP2* at the same time. We aimed to systematically investigate changes in the expression levels of genes related to feed intake under different energy states.

This study is the first to depict the tissue expression profile of the chicken *GOAT* gene, which is highly expressed in the gastric system and the hypothalamus, similar to the expression pattern in mammals [7]. GOAT is an important enzyme that makes ghrelin function, suggesting that *GOAT* may be a gene of interest regarding poultry feed-intake regulation. Our result regarding the expression of *ghrelin* is similar to that reported in other studies; ghrelin is mainly expressed in the proventriculus of chickens, as well as in other tissues (at relatively high expression levels) [30]. Interestingly, in the tissue expression profiling experiments, we did not find changes in *ghrelin* gene expression and serum contents before and after the diet, a finding that was similar to the findings of Richards et al. [45]. However, some researchers have claimed that chicken *ghrelin* level changes with diet in the liver and the small intestine [12,30]. This may be due to the different ages of the experimental animals; they used adult chickens, whereas we used young chicks in this study. Chicken ghrelin may have different sensitivities to diet at different ages, and this requires further research. Although the serum samples were stored at −80 °C until used, we could not avoid the potential degradation of ghrelin, which may be another reason we were unable to detect any differences. As predicted, chicken *GHSR* is widely expressed, with the highest expression level in the advanced central nervous system (including the hypothalamus and the pituitary). Surprisingly, *GHSR* was highly expressed in the jejunum, which is a part of the small intestine. To the best of our knowledge, this study is the first to report a high chicken *GHSR* expression level in the jejunum. The high expression level of *GHSR* in the jejunum may be related to motility in the intestine [46,47]. This will provide a basis for future research on jejunum motility.

LEAP2 is known to antagonize GHSR and influence feed intake in mammals [20,48]. However, its antagonistic function in other non-mammalian vertebrates has not been well explored. A fish experiment demonstrated that LEAP2 serves as an endogenous antagonist of GHSR [49]. Chicken LEAP2 was first identified as an antibacterial peptide [26,50]. Numerous studies have reported the antibacterial effect of chicken LEAP2 on various bacteria or viruses [22,23,24,25,26,27]. In this study, we deemphasized the antibacterial peptide function of chicken LEAP2 and investigated its function as a feed-intake regulator. The gene expression results showed that *LEAP2* responded well to energy changes and was highly expressed in the liver and the small intestine (including the duodenum, jejunum, and ileum); the expression level in the duodenum was the highest. After feeding, the expression level of *LEAP2* was immediately restored to the expression level at the ad libitum state. This indicated that chicken *LEAP2* may function as a potential feed-intake regulator. In terms of gene expression, *LEAP2* responded to energy changes extremely quickly, and the speed was much higher than that of *GHSR* and *ghrelin*, which further revealed the important role of *LEAP2* in feed-intake regulation. The expression level of *LEAP2* decreases after the challenge with a different pathogen or virus (for example, *Eimeria* and Marek’s disease virus) [23,24,25]. This effect may be caused by weakened appetite after the pathogenic infection, or perhaps *LEAP2* has dual functions of antibacterial and energy metabolism simultaneously? Further research is required in this regard. If there is a dual function, chicken *LEAP2* could have good application prospects in the poultry industry. Furthermore, this study reminds us that more attention should be paid to the feed-intake-regulatory function of chicken *LEAP2* in future research.

In addition, the expression levels of *LEAP2* in the duodenum, jejunum, ileum, gizzard, crop, colon, and cecum, which together constitute the digestive tract, were significantly lower than those in the fed and refed groups, which revealed that *LEAP2* may have a similar regulatory mechanism in the digestive tract. Furthermore, in a previous study, researchers found through in situ experiments that *LEAP2* is specifically expressed in jejunal epithelial cells [23], but its regulation is poorly explored. In this study, we explored this specific transcription factor. The prediction results of the transcription factors showed that CDX2 may be the key transcription factor that regulates gene expression in the small intestine (Appendix A). CDX2 is an important intestine-specific transcription factor that regulates intestine-specific gene expression [51]. TTTAT/C is the consensus sequence for the CDX2 binding-site consensus sequence [52]. Chicken *CDX4* is the homologous gene of *CDX2*; the dual luciferase assay of the co-transfection with the chicken CDX4 overexpression vector and the chicken *LEAP2* promoter region pGL3 vector proved that CDX4 could bind to the promoter of *LEAP2*. The conserved sequence TTTAT/C exists in the upstream of *LEAP2* and functions well with CDX4. This also provides a new perspective on the regulation of feed intake and requires further research in the future.

## 5. Conclusions

In conclusion, our gene expression profile (*GOAT*, *ghrelin*, *GHSR*, and *LEAP2*) analysis showed that *LEAP2* is a more flexibly changed feed-intake-related gene that is mainly expressed in the liver and the whole small intestine (including the duodenum, the jejunum, and the ileum) under different energy states. Chicken *GOAT* and *ghrelin* are highly expressed in the proventriculus, and *GHSR* is highly expressed in the hypothalamus, pituitary, and jejunum. Furthermore, this study showed that chicken CDX4 is a potential regulator of *LEAP2* that is highly expressed in the small intestine. The present study highlights the potential role of chicken *LEAP2* in the regulation of feed intake and provides new insights into *LEAP2* regulation.

## Figures and Tables

**Figure 1 animals-12-03496-f001:**
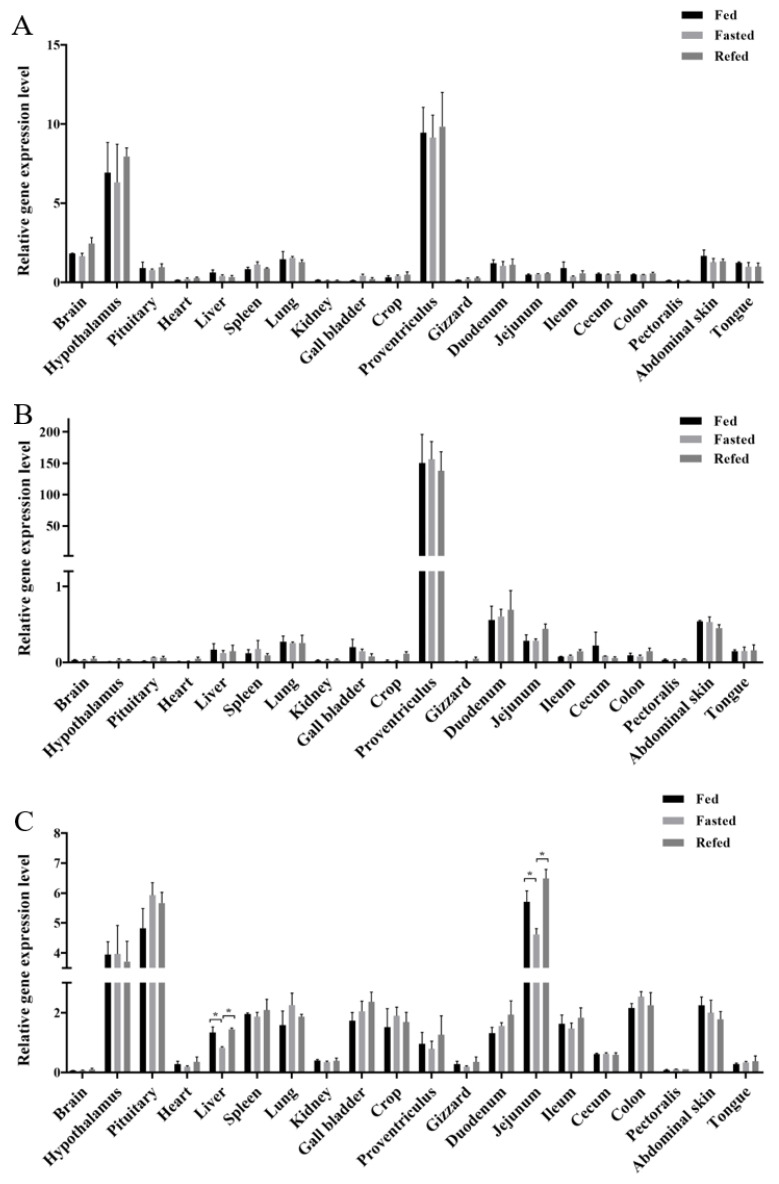
Relative expression levels of *GOAT* (**A**), *ghrelin* (**B**), and *GHSR* (**C**) in the 20 tested tissues of chicks with different energy states (fed, fasted, and refed). *GAPDH* and *HPRT* were used as housekeeping genes to calculate the relative gene expression level. Error bars indicate the SE (n = 6). * indicates significant differences (*p* ≤ 0.05). Abbreviations: *GOAT*, ghrelin O-acyltransferase; *GHSR*, growth hormone secretagogue receptor.

**Figure 2 animals-12-03496-f002:**
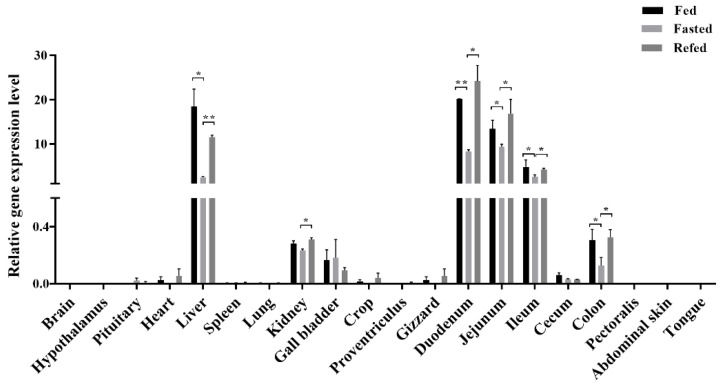
Relative expression levels of *LEAP2* in the 20 tested tissues of chicks with different energy states (fed, fasted, and refed). *GAPDH* and *HPRT* were used as housekeeping genes to calculate the relative gene expression level. Error bars indicate the SE (n = 6). * indicates significant differences (*p* ≤ 0.05), ** indicates significant differences (*p* ≤ 0.01). Abbreviation: *LEAP2*, liver enriched antimicrobial peptide 2.

**Figure 3 animals-12-03496-f003:**
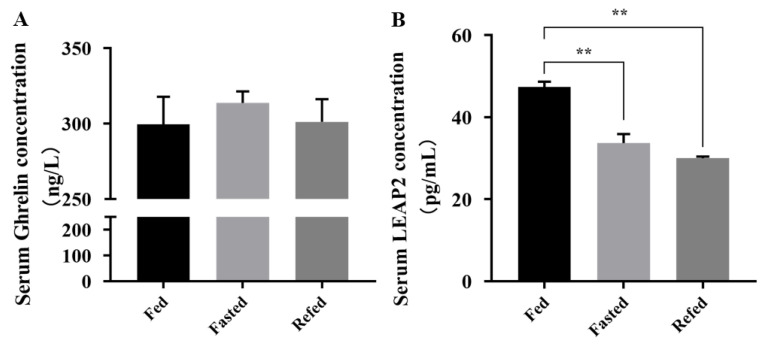
Serum concentrations of ghrelin (**A**) and LEAP2 (**B**) in chicks under different energy states (fed, fasted, and refed). Error bars indicate the SE (n = 6). ** indicates significant differences (*p* ≤ 0.01). Abbreviation: LEAP2, liver enriched antimicrobial peptide 2.

**Figure 4 animals-12-03496-f004:**
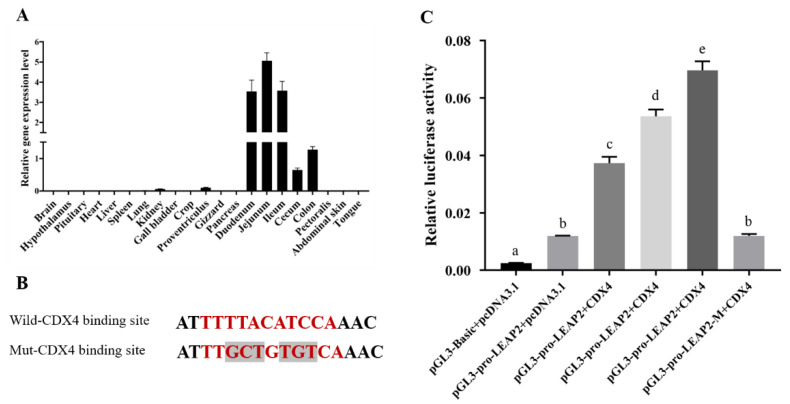
Dual luciferase assay of the 5′ flanking region of chicken LEAP2 gene. (**A**) chicken *CDX4* gene expression profile; (**B**) the strategy to mutate the predicted CDX4 binding site—the red sequence is the predicted CDX4 binding sequence of *LEAP2* and the mutated bases are in black; (**C**) dual luciferase assay of the promoter region of chicken *LEAP2*—the concentration of the overexpression vector group was 1 μg, 2 μg, and 3 μg, and the CDX4 overexpression of the mutated group was 1.5 μg. Different lowercase letters in the bar indicate a significant difference among different groups (*p* ≤ 0.05, n = 3). Abbreviation: CDX4, caudal type homeobox 4.

**Table 1 animals-12-03496-t001:** Parameters of primers used for real time quantitative polymerase chain reaction.

Primer Names	Primer Sequence (5′→3′)	GenBank Number	Size (bp)	Anneal Temperature (°C)
*Ghrelin*-F *Ghrelin*-R	TTTGAAGCACTGCCTAAAGAA GTCATCTTCTCCCTCTGTTTCAT	CGNC: 6373	229	58
*GOAT*-F *GOAT*-R	GACCTGCTCATCCTTCTCCCT TGAGAAGCAGCGTGGCATAA	CGNC: 8705	181	58
*GHSR*-F *GHSR*-R	CATCATCAGGGACAAGAACAAC AAGGCAACCAGCAGAGTATGA	CGNC: 6983	83	58
*LEAP2-*F *LEAP2-*R	TTCTGAGACTGAAGCGGATGA AGGCCGTTCTAAGGAAGCAG	CGNC: 5360	132	58
*CDX4*-F *CDX4*-R	CTCACCCACCAACCAAAGG CAGGGAACTTGTTATTTCATTAGG	CGNC: 49264	352	58
*HPRT*-F *HPRT*-R	CCCAAACATTATGCAGACGA TGTCCTGTCCATGATGAGC	CGNC: 4576	66	58
*GAPDH*-F *GAPDH*-R	CTCTGTTGTTGACCTGACCT CAACCTGGTCCTCTGTGTAT	CGNC: 49077	125	58

## Data Availability

The original data in this article can be obtained directly from the corresponding author. The data are not publicly available due to privacy restrictions.

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
