# Peer review of "Chicken LEAP2 Level Substantially Changes with Feed Intake and May Be Regulated by CDX4 in Small Intestine"

_animals, 2022, doi:10.3390/ani12243496_

Round 1

Reviewer 1 Report

In general, the author abandoned the antibacterial peptide function of chicken LEAP2, and revealed it functional as a feed intake regulator in the paper. This manuscript is well organised and provide sufficient information. However, in some parts of the manuscript can be further modified.

Q1: Line35-37: Strong conclusion, the author should carefully comb and weaken the conclusion.

 Q2: Line 25: it has two consecutive “the”.

 Q3: Line 103: “Plates” should be “plates”.

 Q4: Line 132: “Primerstar max GXL (TaKaRa)”, as far as I know, TaKaRa has no this product, please ensure the name of the product.

Q5: Line 148: gene names should be italicized, please unify other parts of the article one by one.

 Q6: Line 166: “These experiments were repeated three times.” The author should clarify whether to repeat three times in biology or three times in technology.

Q7: Line 168: it should clarify the q-PCR data were collected and analyzed. 

Q8: Line 231: The first letter should be the same as the preceding text, and should be capitalized.

Reviewer 2 Report

Dear Authors, Even though I find merit in your study, a few major concerns must be addressed carefully. The most important factor is the experimental design, for instance, the selection of animal numbers, sample collection, and sample analysis procedures. Thus, correct them. Then, please carefully revise the statistical analysis section so anybody can understand. In addition, I can see many spelling mistakes, unnecessary combining and breaking sentences, and errors in placing spaces between words. Next, please include p-values within the text when you present results, as some results you are talking about can not be clearly understood just by looking at the figures or tables. These mistakes make reading this paper uninteresting to the reader. So, please pay attention to those errors in the future. Thank you.   

Reviewer 3 Report

General points about the manuscript: The manuscript brings interesting information regarding feed intake regulation of chicken. In my opinion the manuscript has a good aim, is of high relevance with new information regarding the activity of different hormones affecting feed intake, however it has some constraints that should be sorted out. The manuscript is poorly written and definitely requires English proofreading in order to improve its readability and comprehension. Additionally, the methodological approach regarding the amount of replicates per treatment is also of great concern and requires further explanation from the authors. Treatments are poorly described in the methodology section and require better description.

Specific considerations:

Title: Avoid abbreviations in the title.

“…may regulated…”, this is grammatically incorrect. Should it be “…may be regulated…”?

Simple summary: Abbreviations should be defined first time mentioned.

Abstract: Abbreviations should be defined first time mentioned.

L13: Sentence requires better transition.

L13: “…is still remains unexplored…” requires revision.

L14: “…aimed to outlined…” requires revision.

L19: “…reported play…” requires revision.

L21: “…Ghrelin was reported has the depressed…” requires revision.

L23: Requires revision.

L25: Delete one of the double “the”.

From this moment on, I will not point anymore what requires revision. The entire text should go through English revision and polishing. I’ll focus the attention on the content of the manuscript, but English revision is definitely required for this manuscript.

Include in abstract a sentence or two regarding the methodology, treatments and experimental design of this study.

L36: “appetite” is not an appropriate scientific word here. “feed intake” instead?

Keywords: For indexing reasons, do not use as keywords those words already mentioned in the title.

Introduction: Define (spell out) the abbreviations used. Abbreviations should be checked throughout the manuscript.

L49: What does it mean “avian specials”?

L59: Incomplete reference.

There are abbreviations in italic and non-italic. Why? Be consistent. Check throughout the manuscript.

L78-84: These sentences do not belong to Introduction, but Conclusion instead.

L90: How many days old? Is it 18 animals of 15 days of age? If they are 15 days old, they are not roosters yet. Please check the information and write the text accordingly.

Was this experiment performed with 6 replicates per treatment? If so, the quality of the day is rather weak. Please provide further explanation and a reason for this in methodology. Experiments in broiler research is performed with many birds in one cage or pen that represents one replicate, however one single animal cannot be used as one replicate.

L92: For methodology use verbs in the past tense.

L93: free for water?

L96-98: There are 19 tissues here, however in Figure there are 20. Please check.

How long did the treatments last? What was the diet of the animals? Treatments are poorly explained. Please improve.

L101-102: Definitely requires revision and improvement.

L142: “constructed” is misspelled.

L144: “descripted”? Isn’t “described”?

L154: Does it mean carbon dioxide? If so, put number 2 subscripted.

L156: Indicate the country as well.

L156: “approcimately” is misspelled. Make sure this manuscript is check for spelling and grammar.

L166: Provide a reason why these experiments were repeated 3 times.

L176: Why 2 P-values? Please provide an explanation.

Results: Include only results and do not include discussion yet. Verbs in results section should be in the past tense.

L186: Revise and rewrite this: “…lung are other tissues have the expression…”

L187: Do not make the assumption “obviously” in results, otherwise, if the result is obvious, why to study it?

Figure 1: Insert in the Y-axis the parameter it refers to. The same in Figure 2.

L218-220: This is not a results sentence, but rather a methodology sentence maybe.

L232-233: Do not insert discussion and reference in the results section. Check throughout the manuscript.

L235: JASPAR database, this is regarding methodology part, not results. Check throughout the manuscript.

L247: does effect?

L253: Mention black, instead of dark.

Discussion: It requires major English revision in order to improve the readability and understanding of its meaning.

Instead of repeating results, please provide a discussion regarding the results.

L310: Be clearer specifying what kind of challenge.

L341: Does Figure S1 matches the Figure S1 mentioned in the text? Also, does it match the uploaded Figure S1?

L343: Table S2 was not mentioned in the text.

Best regards.

Round 2

Reviewer 2 Report

Dear Authors, Thank you for the extensive revision that you provided. It is well improved. 

Reviewer 3 Report

Dear authors,

Thank you for providing a revised version of the manuscript entitled "Chicken LEAP2 Level Substantially Changes with Feed Intake and may be Regulated by CDX4 in Small Intestine". 

I clearly see that the manuscript has gone through an intense revision process that improved its quality. I'm favourable to the acceptance of this manuscript in case my decision is also in line with the Editorial Office of Animals - MDPI.

Regards.